# WristMIR: Coarse-to-Fine Region-Aware Retrieval of Pediatric Wrist Radiographs with Radiology Report-Driven Learning

**Mert Sonmezer** 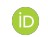     MERT.SONMEZER@METU.EDU.TR
[1] *Department of Computer Engineering, Middle East Technical University, Ankara, Turkiye 06800*

**Serge Vasylechko**[2,3]   SERGE.VASYLECHKO@CHILDRENS.HARVARD.EDU
**Duygu Atasoy**[2,3]      DUYGU.ATASOY@CHILDRENS.HARVARD.EDU
[2] *Quantitative Intelligent Imaging Laboratory, Department of Radiology, Boston Children's Hospital, Longwood Avenue, Boston, MA 02115*
[3] *Harvard Medical School, Longwood Avenue, Boston, MA 02115*

**Seyda Ertekin**[*1,4]      SERTEKIN@METU.EDU.TR
[4] *AI & Big Data Analytics Laboratory, METU-DTX Research Center, Middle East Technical University, Ankara, Turkiye 06800*
**Sila Kurugol**[*2,3]      SILA.KURUGOL@CHILDRENS.HARVARD.EDU

**Editors:** Accepted for publicatin at MIDL 2026

## Abstract

Retrieving wrist radiographs with analogous fracture patterns is challenging because clinically important cues are subtle, highly localized and often obscured by overlapping anatomy or variable imaging views. Progress is further limited by the scarcity of large, well-annotated datasets for case-based medical image retrieval. We introduce **WristMIR**, a region-aware pediatric wrist radiograph retrieval framework that leverages dense radiology reports and bone-specific localization to learn fine-grained, clinically meaningful image representations without any manual image-level annotations. Using MEDGEMMA-based structured report mining to generate both global and region-level captions, together with pre-processed wrist images and bone-specific crops of the distal radius, distal ulna, and ulnar styloid, WristMIR jointly trains global and local contrastive encoders and performs a two-stage retrieval process: (1) coarse global matching to identify candidate exams, followed by (2) region-conditioned reranking aligned to a predefined anatomical bone region. WristMIR improves retrieval performance over strong vision-language baselines, raising image-to-text Recall@5 from 0.82 % to 9.35 %. Its embeddings also yield stronger fracture classification (AUROC 0.949, AUPRC 0.953). In region-aware evaluation, the two-stage design markedly improves retrieval-based fracture diagnosis, increasing mean $F_1$ from 0.568 to 0.753, and radiologists rate its retrieved cases as more clinically relevant, with mean scores rising from 3.36 to 4.35. These findings highlight the potential of anatomically guided retrieval to enhance diagnostic reasoning and support clinical decision-making in pediatric musculoskeletal imaging. The source code is publicly available.[†]

**Keywords:** Region-aware retrieval, Pediatric Wrist radiography, Contrastive pretraining.

---

[*]Co-shared last authorship.

[†]github.com/quin-med-harvard-edu/WristMIR

## 1. Introduction

Wrist fractures are among the most common pediatric injuries, and detecting and classifying them on radiographs is essential for appropriate management. Interpretation in pediatric patients, however, is challenging. Developmental anatomy, including open growth plates, variable ossification centers, and age-dependent changes in bone morphology, introduces substantial variability that can obscure subtle cortical disruptions and complicate distinguishing normal variants from true fractures. Because fracture appearance evolves with age and varies across patients, access to prior radiographs with similar injury patterns can provide valuable diagnostic context and support more consistent decisions.

Despite this need, large and richly labeled pediatric wrist datasets are scarce, as detailed annotations for fracture type, location, and severity require expert pediatric radiologists and are too time-consuming to scale. This has motivated weakly and self-supervised approaches that leverage naturally occurring signals such as paired radiographs and reports. Contrastive language–image models (CLIP and its medical variants) use anatomical and pathological details in radiology reports to learn joint visual–textual representations without manual labels (Lin et al., 2023; Wang et al., 2022; Zhang et al., 2025a; Johnson et al., 2019). These models have advanced medical image retrieval, classification, and domain-specific vision-language modeling, offering scalability well suited to pediatric imaging.

Retrieving radiographs with analogous fracture patterns is especially valuable in pediatrics, where subtle differences often determine treatment decisions. Prior work in content-based image retrieval shows that similar-case retrieval can support diagnosis, reduce uncertainty, and enhance education (Choe et al., 2021; Qayyum et al., 2017; Müller et al., 2004; Dubey, 2022; Hu et al., 2022). Recent frameworks have sought to improve retrieval accuracy by moving beyond global image representations toward anatomy-aware modeling. Methods such as RadIR and AHIVE leverage radiology reports to learn multi-grained similarity and hierarchical visual concepts aligned with specific anatomical structures (Zhang et al., 2025b; Yan et al., 2024). Complementarily, CheXtriev uses graph-based transformers to explicitly model spatial interactions between anatomical regions and pathological findings in chest radiographs (Akash R. J. et al., 2024). However, retrieval remains challenging when clinically meaningful differences are highly localized and subtle (Şaban Öztürk, 2021; Yan et al., 2018; Zhong et al., 2021; Lee et al., 2023). Two radiographs may appear globally similar, yet differ markedly in fracture type, severity, or region.

Global CLIP-style embeddings often fail to capture these fine-grained cues. Subtle findings such as cortical step-off, buckle deformation, physeal widening, or mild tilt/angulation may occupy small regions and are easily diluted by global pooling. Radiographic projections also introduce bone superimposition, causing small but clinically significant features to be lost in coarse representations. Effective retrieval, therefore, requires integrating global wrist context with fine-grained, anatomy-specific detail. Yet, manual annotation of such details is subjective, time-consuming, and difficult to scale (Abacha et al., 2023; Johnson et al., 2019; Nagy et al., 2022).

To address these challenges, we introduce **WristMIR**, a region-aware retrieval framework for pediatric wrist radiographs. WristMIR leverages dense radiology reports to extract anatomy-specific findings and treats sentence-level similarity as a proxy for image-level similarity. These textual signals are paired with bone-specific crops (distal radius, distal ulna,

and ulnar styloid) to train a contrastive language-image model that learns both global wrist representations and localized bone embeddings. At inference, WristMIR performs two-stage retrieval: a global search to identify clinically plausible candidates, followed by region-conditioned reranking for the specified bone. Our main contributions are as follows:

1. **Annotation-free supervision** enabled by a scalable preprocessing pipeline that structures radiology reports into anatomy-specific findings and pairs them with detector generated, bone-level image crops, thereby eliminating the need for manual image annotation, a critical bottleneck in pediatric datasets.

2. **A region-aware representation learning** through a contrastive framework that aligns global wrist images with localized bone representations, enabling fine-grained discrimination of subtle fracture patterns that global embeddings fail to capture.

3. **WristMIR, a two-stage, region-conditioned retrieval framework** that improves clinical relevance over global-only retrieval by first ensuring global compatibility (laterality, morphology) and then refining retrieval based on the local anatomical region.

## 2. Data Preprocessing

Our preprocessing pipeline (Fig. 1) converts wrist radiographs and metadata into training-ready datasets for global and region-aware retrieval by (i) standardizing inputs, (ii) extracting structured anatomy-specific findings from radiology reports, and (iii) producing bone-specific crops and captions for CLIP-based training.

### 2.1. Data Sources

We retrospectively collected 7540 Posterior Anterior (PA) view pediatric wrist radiography examinations from our institution's database under an approved IRB protocol. Each exam is paired with a free-text radiology report authored by board-certified pediatric radiologists. These radiograph-report pairs provide the sole supervision for WristMIR. No manual image-level annotations were used; all labels, regional descriptors, and fracture characteristics are automatically derived through our report-mining pipeline (see § 2.3). We focus on PA views because lateral and oblique projections stack the radius and ulna along the imaging axis, preventing reliable bone-level localization and obscuring fracture cues, making them unsuitable for region-aware retrieval. A full breakdown of dataset composition and fracture distribution is provided in Appendix B.

### 2.2. Wrist-ROI Extraction & Bone Detection

**Detection and cropping.** To isolate relevant anatomical regions, we use YOLOv11-based detectors for wrist–ROI and bone-level localization (Fig. 1a) (Jocher et al., 2023). The ROI detector extracts the primary diagnostic area and removes non-informative regions, achieving a precision of 0.991 and recall of 0.975. The bone-level detector identifies the distal radius, distal ulna, and ulnar styloid with precision 0.947 and recall 1.000. These detected regions are cropped and paired with anatomy-specific captions to construct the

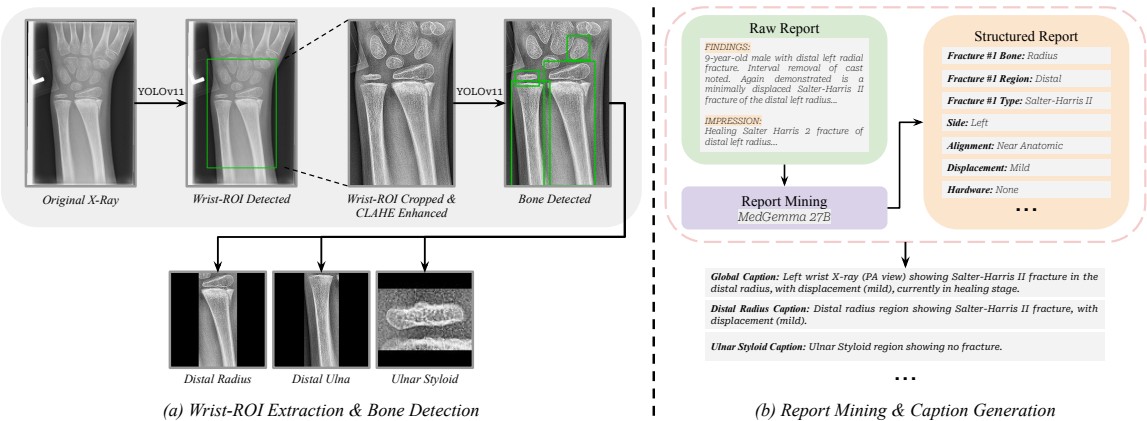

Figure 1: **Data preprocessing pipeline.** (a) YOLOv11 detector first identifies the wrist region of interest (ROI), applies CLAHE enhancement, and then localizes and crops three anatomical regions (distal radius, distal ulna, and ulnar styloid). (b) MEDGEMMA-27B converts each radiology report into a structured representation capturing anatomy-specific findings, which are then used to generate global exam-level captions and region-specific captions aligned with each bone crop.

image–text pairs for region-aware contrastive learning. Detailed performance metrics for each anatomical class and a description of the fine-tuning protocol of textscYOLOv11 are provided in Appendix G.

**Enhancement and padding.** To standardize appearance and improve visibility, all wrist–ROI crops are processed with Contrast-Limited Adaptive Histogram Equalization (CLAHE) (Pizer et al., 1990) and unsharp masking. These steps normalize contrast, enhance cortical boundaries, and reduce variation due to exposure or sharpness differences. Aspect ratios are preserved, and zero-padding is applied to the shorter dimension to produce square CLIP inputs while retaining the original geometry.

## 2.3. Report Mining

**VLM-assisted structuring.** We transform free-text wrist radiology reports into structured representations using the medical VLM MEDGEMMA-27B (Sellergren et al., 2025) (Fig. 1b). Reports are normalized to RADLEX terminology (Langlotz et al., 2006) and parsed into a JSON-like schema capturing anatomical entities, localized fracture descriptors, and global findings (Vasylechko et al., 2025). To reduce hallucinations and enforce schema adherence, we use chain-of-thought prompting with curated examples and validate outputs using PYDANTIC (Colvin, 2017). Post-processing canonicalizes terminology (e.g., "ulna styloid" → "ulnar styloid") and ensures consistency. Manual review of 250 cases shows a hallucination rate below 1 %, indicating robust performance at scale.

**Caption generation.** Each structured report is converted into (i) global captions summarizing the entire wrist examination, including projection, alignment, fracture characteristics, and (ii) region-specific captions describing findings for each anatomical structures (distal

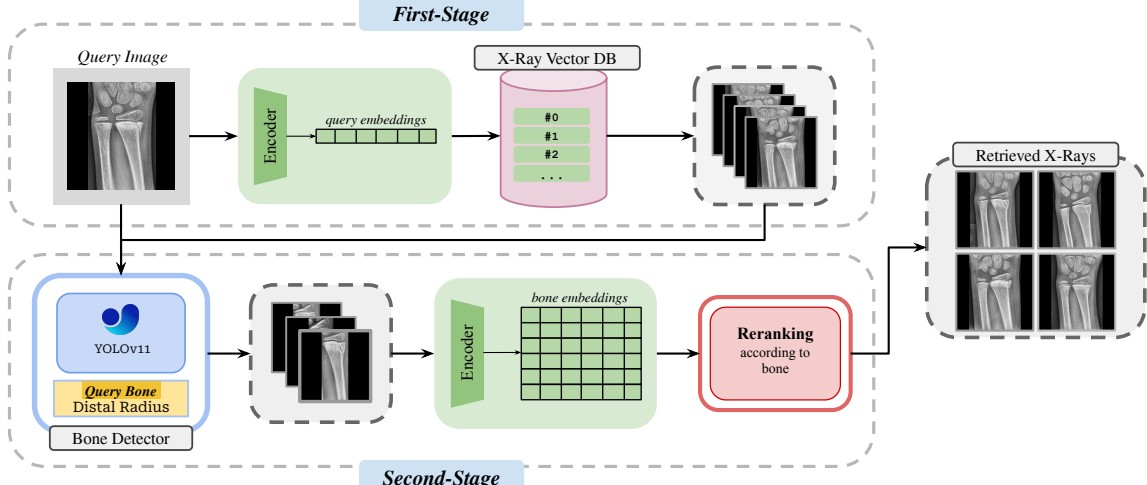

Figure 2: **WristMIR architecture.** A query wrist radiograph is encoded to generate both global and bone-level embeddings. A YOLOv11 detector identifies the relevant bone regions (e.g., distal radius, distal ulna, ulnar styloid). The global embedding is used to retrieve the top-$k$ most similar exams from a precomputed database, after which these candidates are reranked using the region-specific embeddings to enable fine-grained, anatomy-aware retrieval.

radius, distal ulna, or ulnar styloid). Captions are generated via deterministic templates for consistency and serve as text inputs for contrastive training; full assembly logic and examples are provided in Appendix H.

## 3. Methodology: WristMIR

**WristMIR** is a region-aware two-stage retrieval framework (Fig. 2) that learns multi-granular visual-textual representations of pediatric wrist radiographs. The method consists of two components: (i) contrastive learning of global and region-specific embeddings and (ii) region-aware retrieval guided by anatomical queries.

### 3.1. Global and Region-Specific Contrastive Learning

**Architecture.** WristMIR adopts a dual-encoder CLIP framework built on BIOMEDCLIP (Zhang et al., 2025a). The image encoder $\Phi_{\text{img}}(\cdot)$ is a VIT-B/16 and the text encoder $\Phi_{\text{text}}(\cdot)$ is a transformer, both producing 512-dimensional embeddings. For each wrist radiograph $I$ and caption $R$, derived from structured MEDGEMMA-27B reports, the encoders map inputs into a shared embedding space:

$$v = \Phi_{\text{img}}(I), \quad t = \Phi_{\text{text}}(R), \tag{1}$$

aligning paired image–text representations while pushing apart unpaired ones. Training includes both global wrist crops and localized regions (distal radius, distal ulna, ulnar styloid), enabling multi-granular representation learning.

**Training objective.** WristMIR is fine-tuned from BioMedCLIP by unfreezing the last eight image encoder blocks. Each image or crop $I_i$ and caption $R_i$ are encoded as $v_i = \Phi_{\text{img}}(I_i)$ and $t_i = \Phi_{\text{text}}(R_i)$, projected into a shared embedding space.

Because reports often describe normal ("no fracture") examinations, identical or highly similar captions occur frequently, producing ambiguous one-to-one supervision. Wrist radiographs also contain many near-duplicate cases (similar fracture types, healing stages, or regions), making strict single-positive contrastive learning unstable and poorly aligned with the data distribution.

To address this, WristMIR adopts a multi-positive contrastive loss in which all samples sharing the same caption are treated as valid positives. Formally, the symmetric CLIP loss is extended with a positive mask $P_{ij}$ that distributes equal weight over all captions identical to $R_i$:

$$\mathcal{L} = -\frac{1}{B} \sum_{i=1}^{B} \left[ \sum_j P_{ij} \log \frac{\exp(\langle v_i, t_j \rangle / \tau)}{\sum_k \exp(\langle v_i, t_k \rangle / \tau)} + \sum_j P_{ji} \log \frac{\exp(\langle t_i, v_j \rangle / \tau)}{\sum_k \exp(\langle t_i, v_k \rangle / \tau)} \right] \quad (2)$$

where $\tau$ is a learnable temperature. This formulation respects the clinical reality that many examinations convey equivalent semantic information, stabilizes contrastive alignment under limited caption diversity, and enables the model to focus on distinguishing genuinely different fracture patterns rather than arbitrarily separating semantically identical samples. Further analysis and implementation details are provided in Appendices D and C.

## 3.2. Region-Aware Retrieval

**Two-stage retrieval.** WristMIR employs a two-stage retrieval pipeline designed not only for efficiency but, more importantly, to enforce anatomical and view-level consistency before applying fine-grained region analysis. In the *global retrieval* stage, cosine similarity is computed between the query's global embedding and all stored global embeddings:

$$S_g = \langle v_q, v_i \rangle. \quad (3)$$

This produces a candidate pool aligned with the query in coarse clinical properties such as laterality, projection, and wrist morphology. Restricting retrieval to these anatomically consistent cases prevents mismatches (e.g., opposite sides, different projections) and provides a stable basis for downstream region-level analysis.

The second stage performs *region-conditioned reranking*. Given a clinician-specified anatomical region (e.g., distal radius), similarity is computed between the corresponding region-level embeddings:

$$S_r = \langle v_{q,c}, v_{i,c} \rangle, \quad (4)$$

enabling the model to focus on subtle, localized morphological cues for the clinician-specified region. This ensures that fine-grained comparisons occur only among globally compatible candidates, improving clinical relevance.

**Efficient region-level reranking.** Although region-level retrieval relies on YOLO-based bone detection, all detections are performed offline. Bone-specific crops are extracted and

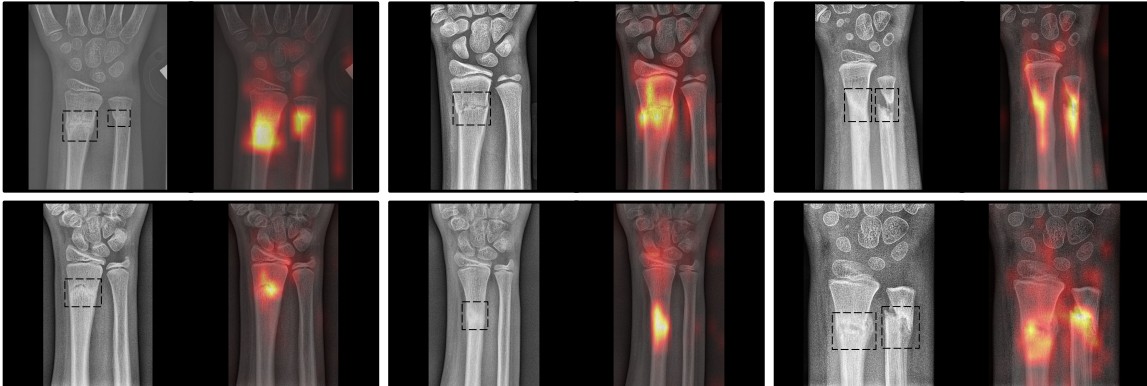

Figure 3: **WristMIR attention maps.** The model consistently attends to fracture-relevant regions, focusing on localized morphological cues. Bounding boxes are shown only to guide visual interpretation of the fracture location and were not included in the dataset or were not used during CLIP training.

encoded once; these indexed embeddings allow reranking to operate on cached representations without query-time inference. To further safeguard against potential detection failures, WristMIR incorporates a fallback mechanism. If the detector fails to localize a specific bone region at inference time, the system automatically reverts to the candidate set generated by the first-stage (global) retrieval. Since the global stage already filters for anatomical consistency, ensuring matching laterality, projection, and morphology, the clinician still receives clinically relevant cases, maintaining system utility even in the absence of fine-grained reranking.

## 4. Experiments

### 4.1. Baselines

We evaluate WristMIR on a pediatric wrist radiograph dataset paired with global and region-specific captions (§ 2). To assess the impact of region-aware learning, we compare against three strong medical vision-language baseline models: BIOMEDCLIP (Zhang et al., 2025a), PMC-CLIP (Lin et al., 2023), and MEDCLIP (Wang et al., 2022). These models represent state-of-the-art CLIP-style approaches pretrained on large biomedical corpora but lack anatomy-specific reasoning.

Additionally, we implement a global-only fine-tuned (GLOBAL-ONLY FT) baseline to isolate the impact of domain adaptation from our region-aware design. This model uses the same configuration as WristMIR but is trained exclusively on global wrist-ROI inputs and global reports, omitting all bone-level crops and region-aware contrastive components. All methodologies are evaluated in a zero-shot setting.

### 4.2. Experimental Setup and Metrics

We conducted all experiments on an evaluation dataset of 876 pediatric wrist images paired with clinical captions, ensuring no overlap with the training set. All retrieval experiments

use a fixed two-stage setup: global retrieval selects the top-100 candidates, followed by region-conditioned reranking to produce the top-10 results. We assess model performance on both zero-shot classification and retrieval tasks using the following metrics:

**Linear Probing.** A logistic regression classifier is trained on frozen image embeddings for binary fracture detection. We report AUPRC, AUROC, and $F_1$ scores to assess the discriminative strength and clinical relevance of the learned visual representations.

**Recall@$k$.** Measures the proportion of queries for which the correct caption appears in the top-$k$ results, evaluating retrieval accuracy for both global and region-level queries.

**Binary fracture & fracture classification matching.** To evaluate diagnostic consistency, we assess whether the top-$k$ ($k = 10$) retrieved cases share the same labels as the query image. For binary fracture matching, a retrieved case is considered a match if its ground-truth binary label (Fracture vs. No-Fracture) is identical to the query. For fracture classification matching, a match requires the specific fracture category (e.g., Salter–Harris, buckle, or transverse) to align. We use a majority voting aggregation to determine the system's final retrieved diagnosis, comparing the benefit of region-aware reranking over single-stage retrieval.

**Radiologist assessment.** A board-certified pediatric radiologist blindly rates the top-$k$ results for diagnostic relevance on a 5-point scale, five indicating the highest relevance.

**Retrieval-based fracture diagnosis.** Fracture presence in each region (distal radius, distal ulna, ulnar styloid) is predicted by aggregating labels from the top-$k$ retrieved cases, and per-region $F_1$ scores are reported.

### 4.3. Unconditional Retrieval and Classification Performance

To assess WristMIR for pediatric wrist classification, we report unconditional image-to-text retrieval and binary fracture classification results in Table 1. WristMIR consistently outperforms all medical CLIP baselines and the global-only fine-tuned model. For a more comprehensive analysis of retrieval quality, including Recall@$k$, Mean Average Precision (mAP), Mean Rank, and Median Rank with with 95 % confidence intervals, see Appendix E.

In image-to-text retrieval, WristMIR achieves higher performance across all $k$ values. It attains a Recall@5 of 9.35 %, compared to 0.82 % for the strongest medical CLIP baseline, BIOMEDCLIP. Given the extreme visual homogeneity of wrist radiographs, where distinct cases often appear globally identical, this 10-times gain reflects meaningful extraction of fine-grained clinical signal. The advantage persists with larger pools: WristMIR reaches a Recall@100 of 52.84 %, nearly doubling the 28.91 % achieved by the GLOBAL-ONLY FT baseline. These gaps highlight that while dataset-specific fine-tuning is beneficial, it does not fully capture the subtle, highly localized morphological cues captured by our multi-granular representation learning. The attention heatmaps in Fig. 3 further support this observation.

Because no existing CLIP model is specialized for wrist fractures, direct retrieval comparisons have inherent fairness limitations. To provide a more balanced evaluation of feature quality, we additionally perform linear probing. This setting assesses the adaptability of

Table 1: **Comparison of WristMIR with medical CLIP baselines and a domain-specific fine-tuned baseline (GLOBAL-ONLY FT).** WristMIR achieves the highest Recall@$k$ across all $k$ values and outperforms all baselines in linear probing metrics, demonstrating stronger wrist-specific representations.

| Method | Recall@$k$ (%) ↑ | | | | Linear Probing | | |
|---|---|---|---|---|---|---|---|
| | $k=5$ | $k=10$ | $k=50$ | $k=100$ | AUROC ↑ | AUPRC ↑ | $F_1$ ↑ |
| MEDCLIP | 0.13 | 0.32 | 1.11 | 2.31 | 0.827 | 0.800 | 0.758 |
| PMC-CLIP | 0.44 | 0.92 | 3.71 | 7.73 | 0.891 | 0.890 | 0.822 |
| BIOMEDCLIP | 0.82 | 1.14 | 5.01 | 10.17 | 0.862 | 0.853 | 0.791 |
| GLOBAL-ONLY FT | 5.83 | 9.41 | 21.71 | 28.91 | 0.898 | 0.913 | 0.815 |
| **Our method** | **9.35** | **15.28** | **38.13** | **52.84** | **0.949** | **0.953** | **0.867** |

embeddings rather than their raw retrieval ability, offering a fairer comparison of representation strength. While medical CLIP baseline models achieve moderate AUPRC scores (0.800–0.890), the GLOBAL-ONLY FT baseline reaches an AUPRC of 0.913 and an $F_1$ of 0.815. WristMIR, in contrast, attains an AUROC of 0.949, an AUPRC of 0.953, and an $F_1$ of 0.867, demonstrating that region-aware contrastive learning produces more discriminative embeddings than global fine-tuning alone.

Despite WristMIR's improvements, absolute retrieval numbers remain modest. This reflects the intrinsic difficulty of the task: unlike classical computer vision retrieval benchmarks, where CLIP distinguishes semantically diverse objects, pediatric wrist radiographs differ only in subtle cortical or physeal abnormalities that are easily obscured by overlapping anatomy. The low absolute values thus reflect task complexity rather than model underperformance, underscoring the challenge of fracture-conditioned retrieval in highly homogeneous medical imaging domains.

## 4.4. Region-Aware Retrieval Evaluation

Table 2 and Fig. 4 compare our two-stage retrieval strategy with a single-stage global baseline. Across all regions and metrics, two-stage retrieval provides consistent improvements. Additionally, a detailed performance comparison between the two-stage strategy and direct region-based retrieval is provided in Appendix F.

The gains are most pronounced for the ulnar styloid, which contains the subtlest fracture patterns. Here, the two-stage strategy improves Binary Fracture Matching from 0.374 to 0.522 and Fracture Classification Matching from 0.344 to 0.468, indicating that global embeddings alone miss subtle, region-specific cues, whereas region-aware refinement successfully emphasizes localized morphological cues.

The qualitative results further support this finding. For ulnar styloid, the two-stage system retrieves cases that closely match the query's fracture type, orientation, and local patterns, whereas the single-stage baseline often surfaces images that are globally similar in projection but pathologically mismatched. This demonstrates that reranking constrains retrieval to anatomically relevant evidence rather than broad global similarity.

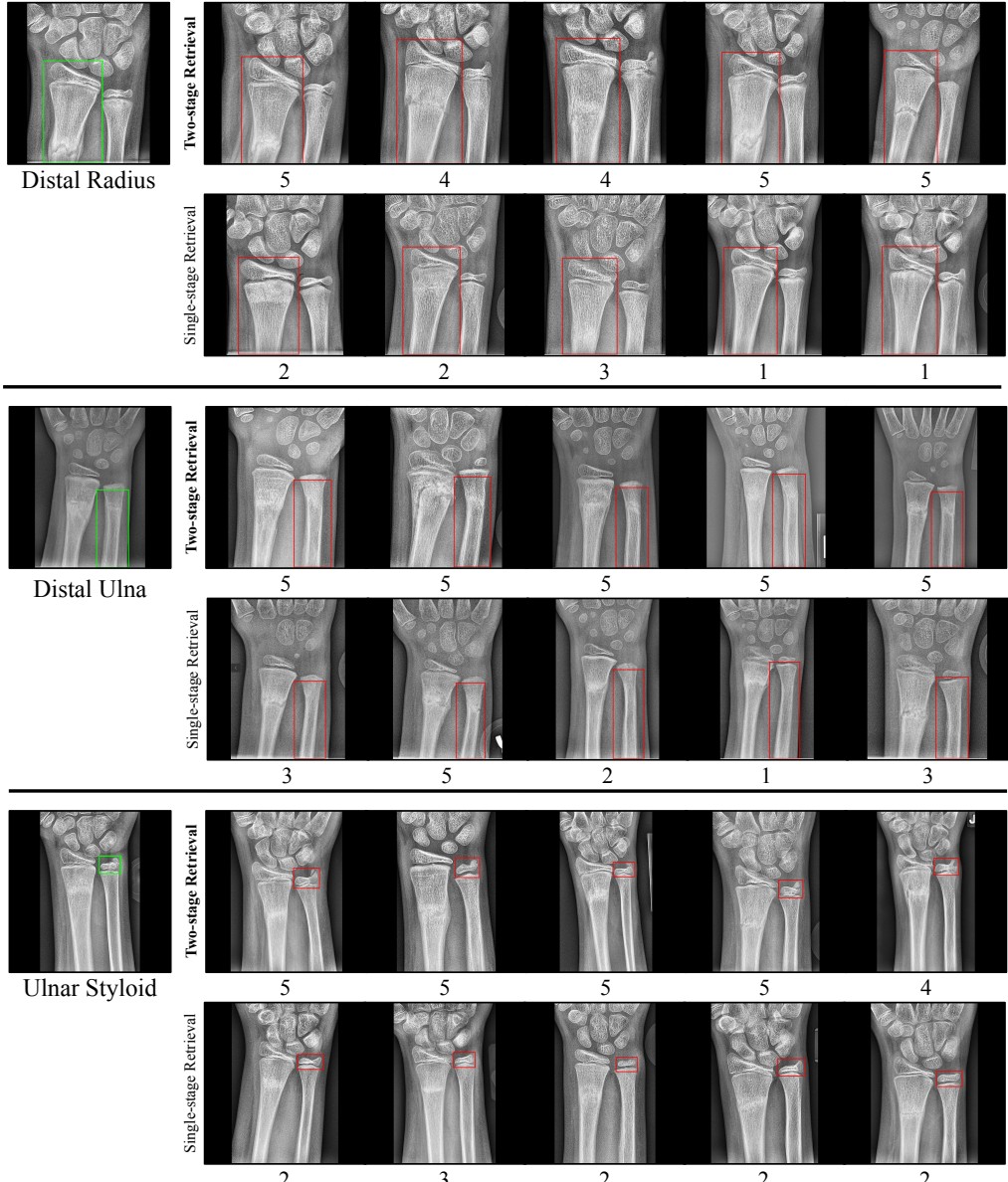

Figure 4: **Comparison of single- and two-stage retrieval.** Region-conditioned rerank-
ing retrieves cases that are anatomically and fracture-pattern aligned, whereas
single-stage retrieval often surfaces globally similar but pathologically mismatched
images. Numbers indicate scores assigned by a pediatric radiologist, showing
higher and more clinically relevant retrieval for the proposed two-stage method.

Radiologist assessments follow the same trend. In a blinded 5-point evaluation on 30
query radiographs (20 retrieved images per query 10 per system), the two-stage method
clearly outperforms the single-stage baseline. For ulnar styloid, scores rise from 3.16 to
4.41, indicating that region-conditioned retrieval produces images experts consider more
diagnostically meaningful. We hypothesize that the single-stage model frequently retrieves

Table 2: **Single- vs. two-stage retrieval.** Retrieval performance across anatomical regions. The two-stage method consistently improves binary fracture matching, fracture classification, fracture diagnosis, and radiologist-rated clinical relevance compared with single-stage retrieval.

| Method | Distal Radius | Distal Ulna | Ulnar Styloid |
|---|---|---|---|
| *Binary Fracture Matching* | | | |
| Single-stage | 0.824 | 0.458 | 0.374 |
| **Two-stage** | **0.864** | **0.666** | **0.522** |
| *Fracture Classification Matching* | | | |
| Single-stage | 0.534 | 0.372 | 0.344 |
| **Two-stage** | **0.578** | **0.542** | **0.468** |
| *Radiologist Assesment* | | | |
| Single-stage | 3.47 | 3.45 | 3.16 |
| **Two-stage** | **4.22** | **4.42** | **4.41** |
| *Retrieval-based Fracture Diagnosis* | | | |
| Single-stage | 0.897 | 0.574 | 0.233 |
| **Two-stage** | **0.934** | **0.771** | **0.554** |

visually similar but clinically irrelevant cases, whereas WristMIR's reranking step elevates candidates with fracture patterns that accurately correspond to the queried anatomy.

### 4.5. Retrieval-based Fracture Diagnosis

We further assess whether region-aware retrieval supports diagnosis by aggregating the region-level fracture labels of the top-$k$ retrieved exams. Table 2 reports per-region $F_1$ scores. The single-stage baseline performs well for the distal radius (0.894) but deteriorates for the distal ulna (0.574) and ulnar styloid (0.233). Incorporating region-conditioned reranking improves performance to 0.934, 0.771, and 0.554, respectively. The largest improvement occurs in the ulnar styloid, where fracture patterns are subtle and easily overshadowed by global appearance, reinforcing that anatomically targeted retrieval better captures subtle fracture patterns.

### 5. Conclusion

We introduced WristMIR, a region-aware retrieval framework for pediatric wrist radiographs that integrates structured report mining with joint global–local representation learning. By using both global and bone-level representations, WristMIR enables efficient, anatomy-specific retrieval of clinically analogous cases, improving diagnostic confidence, treatment planning, and education. Its two-stage retrieval design delivers fine-grained accuracy while remaining computationally practical for real-time use in clinical workflows. More broadly, WristMIR illustrates how structured radiology reports and anatomically grounded reasoning can overcome key limitations of global CLIP-style models for medical images, offering a scalable foundation for next-generation clinical decision support tools. Clinical deployment considerations and methodological limitations are discussed in Appendix A.

## Acknowledgments

This research was supported partially by the National Institute of Diabetes and Digestive and Kidney Diseases (NIDDK) of the National Institutes of Health under Award No. R01DK125561, partially by the Massachusetts AI Hub Award, and by the Research Universities Support Program (YÖK-ADEP) under Project No. ADEP-312-2024-11490. Mert Sonmezer also received scholarship support from Insider One.

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

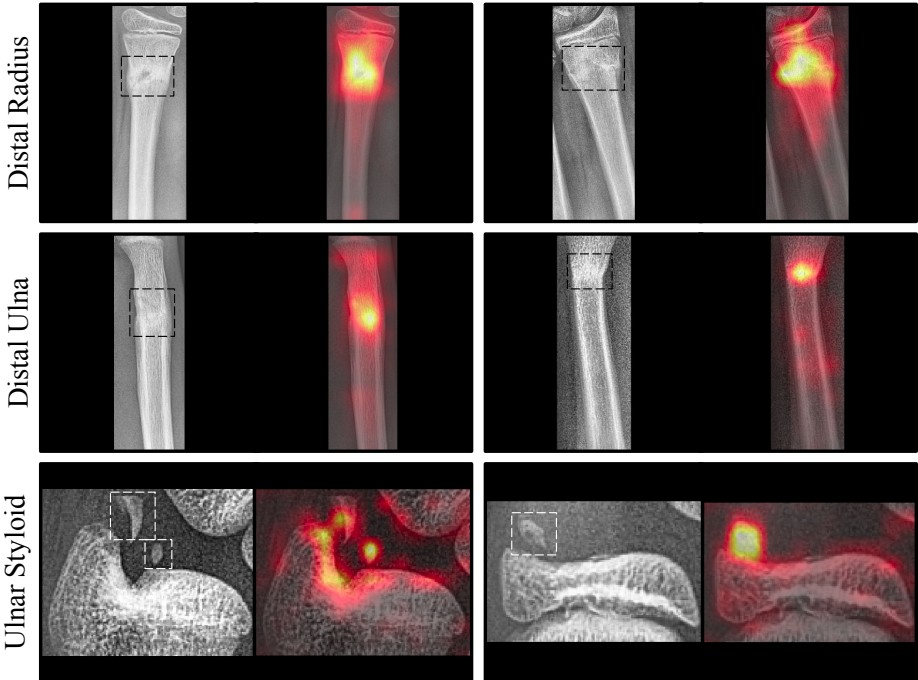

Figure 5: **WristMIR bone-level attention maps.** For each anatomical region (distal radius, distal ulna, and ulnar styloid), the model concentrates its attention on localized morphological cues that align with fracture-relevant structures. The dashed bounding boxes are included only to guide the reader by indicating the approximate fracture locations.

## Appendix A. Limitations and Future Directions

WristMIR relies on detector accuracy and structured report quality; failures in either can affect region-level retrieval. While structured reports standardize clinical semantics, they may omit nuanced context present in free text, making the use of "soft" embeddings from raw reports a promising direction for future work. Future studies should also evaluate classify-then-retrieve baselines to better disentangle the contributions of category prediction and representation learning and to assess how misclassification errors propagate through the pipeline. Additionally, hardware and casts may still bias the global encoder despite our region-aware design. Finally, evaluation is limited to a single institution and does not assess cross-domain generalization; although the two-stage retrieval pipeline is computationally efficient and compatible with clinical workflows, prospective validation is required prior to real-world deployment.

## Appendix B. Dataset Details

Our pediatric wrist radiograph dataset contains 7540 examinations paired with structured radiology reports produced through our VLM-based report-mining pipeline. Across all stud-

Table 3: **Dataset composition and fracture distribution.** Summary of dataset used for WristMIR training and evaluation.

| Dataset Overview | |
|---|---|
| Total examinations | 7,540 |
| Total region-level fractures | 8,637 |
| Normal cases (0 fractures) | 2,209 |
| Fracture-positive cases ($\geq 1$) | 5,331 |
| *Fractures per Case* | |
| 0 | 2,209 |
| 1 | 2,264 |
| 2 | 2,869 |
| 3 | 159 |
| 4 | 37 |
| 5 | 2 |
| *Fracture Location* | |
| Distal radius | 5,369 |
| Distal ulna | 2,030 |
| Ulnar styloid | 1,238 |
| *Fracture Morphology* | |
| Salter–Harris | 1,621 |
| Buckle | 1,007 |
| Transverse | 1,924 |
| Healing fractures | 3,955 |

ies, we identify 8637 region-level fractures. A total of 2209 cases are normal (0 fractures), while 5331 contain one or more fractures. Most fracture-positive cases involve one or two regions, with only a small number involving multiple sites.

Fractures most commonly involve the distal radius (5369 instances), followed by the distal ulna (2030) and ulnar styloid (1238). The dataset includes a diverse range of fracture types, notably 1621 Salter–Harris, 1007 buckle, and 1924 transverse fractures, alongside 3955 healing fractures, providing broad clinical variability useful for training region-specific embeddings. Table 3 includes the key statistics.

## Appendix C. CLIP Training Details

WristMIR's encoders are fine-tuned from BiomedCLIP (PubMedBERT–ViT-B/16) using the OpenCLIP framework (Ilharco et al., 2021) on 4 NVIDIA A100 GPUs. The model is initialized from `microsoft/BiomedCLIP-PubMedBERT_256-vit_base_patch16_224` on HuggingFace, with a ViT-B/16 visual encoder and BiomedBERT text encoder projected into a shared 512-dimensional space. Consistent with our region-aware design, only the final eight visual transformer blocks are unfrozen, and standard CLIP preprocessing (RGB 224×224, bicubic interpolation, mean/std normalization) is applied.

Training uses `AdamW` (`lr`=$1 \times 10^{-5}$, `weight decay`=0.01, $\beta_1/\beta_2$=0.9/0.98) with a cosine schedule and 50 warmup steps over 30 epochs. A global batch size of 2048 (512×4) is used with gradient clipping at 1.0. Because many radiographs share similar report-derived

Table 4: **Impact of multi-positive contrastive loss.** Comparative evaluation of the multi-positive (MP) formulation against the standard single-positive CLIP objective.

| Method | Recall@$k$ (%) ↑ | | | | Linear Probing | | |
|---|---|---|---|---|---|---|---|
| | $k=5$ | $k=10$ | $k=50$ | $k=100$ | AUROC ↑ | AUPRC ↑ | $F_1$ ↑ |
| w/o MP Loss | 9.22 | 15.12 | 38.03 | **53.38** | 0.949 | 0.953 | 0.866 |
| **w/ MP Loss** | **9.35** | **15.28** | **38.13** | 52.84 | 0.949 | 0.953 | **0.867** |

captions, a multi-positive contrastive loss is employed, treating all identical captions as valid positives to better match clinical supervision patterns.

Figure 5 shows additional attention maps from the fine-tuned image encoder, demonstrating WristMIR's improved anatomical specificity across the distal radius, distal ulna, and ulnar styloid. After training, attention consistently localizes to fracture-relevant regions, cortical margins, metaphyseal interfaces, and subtle irregularities, illustrating how multi-granular supervision reshapes the embedding space toward clinically interpretable cues and supports the gains observed in region-aware retrieval.

## Appendix D. Impact of Multi-Positive Contrastive Loss

We compared our multi-positive (MP) formulation (Eq. 2) against a standard single-positive CLIP objective to assess its influence on representation quality. While aggregate metrics remain stable (Table 4), the MP loss is mathematically better aligned with our report-mined supervision. By allowing semantically identical samples to be treated as valid positives, this objective prevents the model from learning artificial, non-clinical features to satisfy a strict one-to-one mapping. Consequently, the MP loss ensures that the resulting embedding space is organized by underlying clinical pathology rather than arbitrary sample indices.

## Appendix E. Expanded Retrieval Metrics

To provide a more comprehensive view of retrieval performance beyond Recall@$k$, we evaluate WristMIR and the baselines using Mean Average Precision (mAP), Mean Rank, and Median Rank, reporting 95% confidence intervals (CIs) to assess statistical significance. As shown in Tables 5 and 6, WristMIR significantly outperforms both medical CLIP and domain fine-tuned baselines across all ranking and recall metrics. Notably, our model achieves a Median Rank of 89 [CI: 83, 97], a more than 5-fold improvement over the GLOBAL-ONLY FT baseline (473 [CI: 439, 512]). Furthermore, WristMIR exhibits an mAP of 7.34% [CI: 6.69, 7.96], representing an 8-fold improvement over the strongest zero-shot baseline, BIOMEDCLIP (0.89% [CI: 0.70, 1.11]). The non-overlapping CIs across these primary metrics confirm that WristMIR consistently pushes relevant clinical cases toward the top of the retrieval list.

Table 5: **Expanded retrieval performance comparison.** Evaluation of ranking quality and result distribution using Mean Average Precision (mAP), Mean Rank, and Median Rank. Values are reported with 95 % confidence intervals in brackets. WristMIR achieves a significantly lower Median Rank and higher mAP, demonstrating its ability to consistently push relevant clinical matches to the top of the candidate list.

| Method | mAP(%) ↑ | Mean Rank ↓ | Median Rank ↓ |
|---|---|---|---|
| MEDCLIP | 0.24 [0.15, 0.34] | 1801.14 [1770.97, 1831.02] | 1874 [1838, 1910] |
| PMC-CLIP | 0.65 [0.52, 0.80] | 886.54 [862.24, 911.27] | 700 [658, 734] |
| BIOMEDCLIP | 0.89 [0.70, 1.11] | 914.68 [888.34, 940.80] | 759 [723, 806] |
| GLOBAL-ONLY FT | 4.41 [3.90, 4.91] | 812.56 [782.37, 843.32] | 473 [439, 512] |
| **Our method** | **7.34 [6.69, 7.96]** | **141.82 [136.70, 147.58]** | **89 [83, 97]** |

Table 6: **Expanded Recall@$k$ performance comparison.** Evaluation of retrieval recall. Values are reported in percentage (%) with 95 % confidence intervals in brackets. WristMIR consistently outperforms baselines across all recall levels.

| | Recall@$k$ (%) ↑ | | | |
|---|---|---|---|---|
| | $k = 5$ | $k = 10$ | $k = 50$ | $k = 100$ |
| MEDCLIP | 0.13 [0.00, 0.25] | 0.32 [0.13, 0.54] | 1.11 [0.73, 1.52] | 2.31 [1.81, 2.85] |
| PMC-CLIP | 0.44 [0.22, 0.70] | 0.92 [0.60, 1.27] | 3.71 [3.04, 4.41] | 7.73 [6.85, 8.68] |
| BIOMEDCLIP | 0.82 [0.54, 1.14] | 1.14 [0.79, 1.55] | 5.01 [4.34, 5.86] | 10.17 [9.16, 11.28] |
| GLOBAL-ONLY FT | 5.83 [5.04, 6.59] | 9.41 [8.34, 10.43] | 21.71 [20.25, 23.14] | 28.91 [27.16, 30.49] |
| **Our method** | **9.35 [8.34, 10.30]** | **15.28 [14.01, 16.51]** | **38.13 [36.35, 39.84]** | **52.84 [51.09, 54.58]** |

## Appendix F. Analysis of Coarse-to-Fine Retrieval Strategy

We assess the importance of the proposed coarse-to-fine design by comparing the two-stage retrieval strategy with a single-stage, region-only approach. As shown in Table 7, the two-stage strategy performs comparably to and, in certain anatomical regions such as the ulnar styloid, exceed direct region-based retrieval. These results indicate that the initial global retrieval stage effectively preserves fracture-relevant cases for fine-grained reranking.

Beyond diagnostic accuracy, our two-stage design is clinically motivated to ensure global anatomical consistency. Relying exclusively on localized regions can result in matches that are anatomically similar but clinically inconsistent regarding laterality, position, and age-dependent morphology. Architecturally, this design ensures high efficiency; while retrieval from a precomputed cache is near-instantaneous, the two-stage approach is essential for scaling to new or non-cached databases where bone detection and embedding extraction must be performed on-the-fly. For such non-cached archives, running detection across the entire database for every query is computationally prohibitive. As shown in Table 7, retrieval latency for non-cached queries scales with the size of the candidate pool. By restricting fine-grained matching to a pre-filtered set ($k = 100$), we achieve a mean retrieval time of 7.81s, compared to 74.94s for a larger pool ($k = 1000$), enabling real-time integration into clinical workflows.

Table 7: **Performance comparison of region-only retrieval and two-stage retrieval along with retrieval time analysis.** Comparison between direct region-based single-stage and two-stage retrieval across binary fracture matching and fracture classification matching.

| Method | Distal Radius | Distal Ulna | Ulnar Styloid |
|---|---|---|---|
| *Binary Fracture Matching* | | | |
| Region-based | **0.892** | **0.670** | 0.516 |
| **Two-stage** | 0.864 | 0.666 | **0.522** |
| *Fracture Classification Matching* | | | |
| Region-based | **0.592** | 0.522 | 0.344 |
| **Two-stage** | 0.578 | **0.542** | **0.468** |

| *Retrieval Time Analysis* | | | |
|---|---|---|---|
| Pool Size $(k)$ | $k = 100$ | $k = 500$ | $k = 1000$ |
| **Mean Time (s)** | 7.81 | 40.39 | 74.94 |

## Appendix G. YOLO-based Bone Localization

To make sure the clinical reliability of the inference pipeline, we evaluated our YOLOv11s bone detector on a held-out validation set of pediatric wrist radiographs. The detector was initialized with COCO pre-trained weights and specifically fine-tuned on a dedicated subset of pediatric radiographs to learn the clinical semantics of the distal radius, distal ulna, and ulnar styloid. This supervised fine-tuning protocol employed a training set of 385 images containing 1.155 manual annotations and a validation set of 42 images with 126 manual annotations. As shown in Table 8, the resulting model achieves exceptional localization performance, reaching 100 % recall across all three anatomical regions. This indicates that the system consistently identifies the regions of interest . The slight reduction in precision for the distal ulna and ulnar styloid stems from rare false positives.

Table 8: **Bone detector performance metrics.** Localization performance of the YOLOv11s model across the three primary anatomical regions of interest. The 100 % recall ensures that no regions were missed in the evaluation set.

| Anatomical Region | Precision | Recall | $F_1$ | mAP@50 |
|---|---|---|---|---|
| Distal Radius | 0.977 | 1.000 | 0.988 | 0.995 |
| Distal Ulna | 0.933 | 1.000 | 0.967 | 0.995 |
| Ulnar Styloid | 0.933 | 1.000 | 0.967 | 0.995 |
| Overall | 0.947 | 1.000 | 0.973 | 0.995 |

## Appendix H.  Automated Caption Generation Templates

For reproducibility, we include the deterministic templates used to transform structured metadata into natural language captions for both global images and localized regions.

### H.1.  Full Image (Global) Template

Global captions are constructed by concatenating anatomical and pathological components into a structured report following a fixed assembly logic:

- **Structure:** [Side] wrist X-ray, [View] view showing [Fracture Details]. [Additional Findings].

- **Example:** *Left wrist X-ray (PA view) showing Salter-Harris fracture in the distal radius, currently in healing stage.*

### H.2.  Region-Specific Template

Region captions focus exclusively on the anatomical area within a specific crop (e.g., distal radius, distal ulna, ulnar styloid) to provide localized supervision for contrastive learning.

- **Structure:** [Region Name] region showing [Fracture Details].

- **Example:** *Ulnar styloid region showing fracture in the ulnar styloid, with displacement (mild).*

