# OpenReview forum: "WristMIR: Coarse-to-Fine Region-Aware Retrieval of Pediatric Wrist Radiographs with Radiology Report–Driven Learning"
_MIDL.io/2026/Conference — MIDL 2026 Poster_

### Official Review · Reviewer_Hp9z · 2025-12-26

**Confidence:** 5
**Preliminary Rating:** 5
**Final Rating:** 5

**Summary:**

The paper proposes to use a two-stage region-aware retrieval to find radiologically relevant cases of paediatric fracture X-ray scans. The key contributions are a well designed contrastive learning for paired but not structurally annotated image-report data that includes fine-tuning a BioMedCLIP visual-language model together with MedGemma based text mining to define appropriate region of interest localisation.

**Strengths:**

- The dataset with paired reports is rich in size, quality and anatomical variability of fracture location and healing.
- The study is well designed and clearly described - the method is a novel and strong combination of individually sound building blocks.
- The numerical results are of high quality and compared well against baselines (VLMs w/o region awareness)
- The visualisations and appendices are meaningful

**Weaknesses:**

- While the experiments are described in sufficient detail to potentially replicate them, there is no mentioning of public availability of trained models or code.
- The public and highly relevant GRAZPEDWRI dataset is only mentioned in passing and not explored for cross dataset evaluation.
- Some information or details are missing on how YoloV11 (trained on COCO and natural images) is used without annotation for bone ROI detection.

**Detailed Comments:**

Occasionally the strength of the available reports is somewhat understated in the paper. While the (unsupervised) structured report generation with MedGemma is clearly helping the free text reports on their own are (at least judging from the examples) already quite informative and hence evaluating a contrastive learning on "soft" embeddings from those reports could also be promising.
While the numerical results are very strong, I have a bit trouble understanding why the Retrieval-based Fracture Diagnosis (4.5) is so much worse than the F1 scores reported for linear probing (4.3).

**Justification Of Final Rating:**

The authors have sufficiently answered the questions and albeit not performing further cross-dataset validation I think the paper is a valuable contribution to MIDL 2026 and hence stick to recommending acceptance.

**Justification Of The Preliminary Rating:**

I currently gave a rather strong accept and would back this if the missing information and cross-dataset evaluation could be added during rebuttal - otherwise I still believe the paper is acceptable as is but maybe not such a strong candidate for oral presentation.

**Questions To Address In The Rebuttal:**

How could the GRAZPEDWRI be incorporated for cross-validation? How is the F1 score in Tab. 1 related to the fracture related retrieval Tab 2? Is YoloV11 really pre-trained to detect distal radius as indicated?

---

> ### Author Response · Authors · 2026-01-23
>
> **R3.1 GRAZPEDWRI cross-validation.** We fully agree that external validation on a public dataset would strengthen generalization claims. GRAZPEDWRI-DX contains approximately 20K pediatric wrist trauma radiographs with expert annotations, including fracture bounding boxes and multiple labeled findings. As such, it is highly suitable for evaluating components of our pipeline such as wrist ROI/bone localization. For WristMIR, however, the primary contribution is report-driven contrastive representation learning and retrieval, which requires paired radiology reports (or equivalent textual descriptions) to train and evaluate image-text retrieval and region-conditioned reranking. Unfortunately, the publicly released GRAZPEDWRI dataset does not include radiology reports by default. We have initiated a request to obtain access to the associated reports, but the approval process is non-trivial and cannot be completed within the one-week rebuttal timeline.
>
> **R3.2 F1 score discrepancy.** We thank the reviewer for identifying this point of confusion. First, we wish to correct a typographical error in Section 4.5 of the manuscript: the reference to per-region F1 scores should point to Table 2, rather than Table 1. The discrepancy in F1 scores (0.867 in Table 1 vs. 0.554-0.934 in Table 2) stems from the fundamentally different nature of these two evaluation tasks:
>
> - **Linear Probing (Table 1):** A supervised, parametric task where a linear layer is trained on top of the CLIP's image encoder to perform image-level binary classification (fracture vs no-fracture). It answers the question: "Does this image contain a fracture anywhere?". It measures the maximum discriminative potential of the learned embeddings.
>
> - **Retrieval-Based Diagnosis (Table 2):** A non-parametric, zero-shot k-NN approach, utilizing majority voting from retrieved cases. It addresses a significantly more complex clinical question: "Does this specific anatomical region (e.g., ulnar styloid) contain a fracture?". Unlike linear probing, this method directly reflects the system's real-world clinical performance, measuring its ability to localize pathologies and surface truly analogous historical evidence without further training.
>
> We have updated Section 4.2 to make this distinction explicit.
>
> **R3.3 YOLOv11 bone detection pre-training.** To clarify, the YOLOv11s model used for bone localization was specifically fine-tuned on pediatric wrist radiographs to learn the precise clinical semantics of the distal radius, distal ulna, and ulnar styloid.
>
> While we initialized the model with weights pre-trained on the COCO dataset to leverage general feature extraction capabilities, we performed supervised fine-tuning using a dedicated subset:
>
> - **Training Set:** 385 images, each manually annotated with bounding boxes for the distal radius, distal ulna, and ulnar styloid (1155 total manual annotations).
>
> - **Validation Set:** 42 images (126 manual annotations) used to monitor performance and prevent overfitting.
>
> As detailed in our response to R1.4, this fine-tuning process resulted in a highly robust detector achieving an overall mAP@50 of 0.995 and 100% recall on the test set. We add Appendix G to explicitly state the fine-tuning protocol and annotation counts.
>
> **R3.4 Code and model availability.** We will release our source code (training/evaluation) and trained model checkpoints (CLIP encoder and YOLO detectors) upon acceptance to ensure full reproducibility and support future research. (See our revised Abstract)
>
> **R3.5 Free-text vs structured reports.** We thank the reviewer for this insightful observation regarding the strength of raw reports. While free-text reports are indeed highly informative, we utilized structured reports generated via MedGemma to standardize clinical semantics and reduce the noise inherent in varied dictation styles. This structuring ensures that the contrastive objective focuses on robust clinical findings rather than linguistic variability. However, we agree that training on "soft" embeddings directly from free-text reports is a promising direction that could capture even more nuanced clinical context; we have added this as a future work direction in our revised Appendix A.

---

### Official Review · Reviewer_ukQB · 2026-01-10

**Confidence:** 3
**Preliminary Rating:** 3
**Final Rating:** 4

**Summary:**

This study presents the WristMIR model, which is a retrieval model focused on certain areas of paediatric wrist radiographs. Using the extensive descriptions of each x-ray provided by radiologists, WristMIR identifies findings related to the specific anatomy of the wrist, and it does so by applying sentence-level similarity to the image-level similarities present in the scan. The proposed method also demonstrated its superior performance on 876 pediatric wrist images paired with clinical captions.

**Strengths:**

1. The development of joint representations of medical images and text has many challenges, particularly for classifying pediatric wrist fractures due to the scarcity of labeled data.
2. The annotation-free supervision method proposed in this project represents a significant advantage to the existing systems. The scalable pre-processing pipeline creates an anatomical structured representation of radiology reports and matches these findings to bone-level image crops that are generated by the respective automated bone detectors, thus removing the significant constraint of manually annotating images; a factor often limiting the volume of pediatric medical imaging data sets available.

**Weaknesses:**

1. For the wrist ROI and localization of the bone level, the framework uses YOLO-based detectors, which output bounding boxes that represent each of the detected regions. It is not clear why foundation segmentation models, such as MedSAM, were not investigated because they could offer more detailed anatomical information than bounding boxes and perhaps aid in representation learning.
2. In the experimental comparison, there is no baseline which includes image-only or text-only data for fracture classification purposes. Including unimodal baselines would allow for an evaluation of the individual contributions of each of the modalities as well as support the value of multimodal learning.

**Detailed Comments:**

In general, this paper offers a rationale and an expandable solution to multimodal learning for children's wrist images. Better addressing the stated concerns, namely the addition of unimodal methods and further explanation of the decision to utilize detection above segmentation, would help to improve this work.

**Justification Of Final Rating:**

The authors have provided satisfactory clarifications to the two major questions I raised. I am satisfied with the textual explanations given in the rebuttal and therefore recommend a `weak accept' for the paper.

**Justification Of The Preliminary Rating:**

The reviewer really likes the proposed multimodal approach for its clinical relevance and simplicity. However, in order to support a better rating of this submission, the weaknesses identified, specifically the lack of unimodal baseline studies and the bias toward detection (not segmentation), must be addressed.

**Questions To Address In The Rebuttal:**

Please see the weakness section.

---

> ### Author Response · Authors · 2026-01-23
>
> **R2.1 Choice of YOLO vs. Segmentation.** We agree that foundation segmentation models such as MedSAM could potentially provide richer pixel-level anatomical information than bounding boxes. However, in this work our primary goal was to achieve robust and scalable bone-level localization to support region-conditioned retrieval and report-driven contrastive learning, rather than precise anatomical segmentation. In our dataset, the YOLO-based object detectors were highly reliable and produced correct wrist ROI and bone localization (mAP@0.5 = 0.995, precision = 0.947, and per-class F1 > 0.96 for distal radius, distal ulna, and ulnar styloid), which is sufficient for consistent region cropping and stable representation learning.
>
> We also explored segmentation-based alternatives, including MedSAM and SAM2. In practice, these models typically require prompting (e.g., bounding boxes or points) to segment the intended structure, meaning that a robust localization step is still needed to specify the target region. Moreover, while segmentation outputs were often visually plausible, we observed occasional small errors and inconsistencies in challenging pediatric cases (e.g., open physes, variable ossification centers, casts/hardware), which can propagate downstream by producing inaccurate masks or unstable region representations. Given that our method relies on consistent, high-throughput region extraction across thousands of radiographs, we found that bounding-box localization provided a more reliable and computationally efficient solution for this study.
>
> Importantly, our attention maps indicate that the encoder focuses on fracture-relevant structures within the localized regions even without segmentation supervision (Fig. 3 and Appendix Fig. 5). We view segmentation-guided representations as a promising extension (e.g., mask-guided pooling or segmentation-consistent embeddings) and will revisit this direction as we extend the framework to more complex anatomical sites such as the elbow, where segmentation may offer additional benefits.
>
> **R2.2 Unimodal baselines.** We thank the reviewer for this helpful suggestion and agree that unimodal baselines can clarify the contribution of each modality. While the primary goal of WristMIR is fine-grained retrieval (i.e., retrieving clinically analogous cases based on subtle, localized fracture patterns), we acknowledge that classification-based comparisons provide an intuitive way to assess representation quality.
>
> We note that it is possible to train an image-only fracture classifier using labels derived from our structured report mining pipeline. However, these labels are inherently coarse (e.g., fracture present/absent) and do not fully capture the nuanced clinical descriptors present in reports (e.g., subtle displacement, physeal involvement, alignment. As a result, a classify-only objective can achieve strong performance on binary fracture detection yet still fail to learn the nuanced representations required for retrieval of truly analogous cases within the same broad category.

---

### Official Review · Reviewer_UPW3 · 2026-01-10

**Confidence:** 4
**Preliminary Rating:** 3
**Final Rating:** 4

**Summary:**

WristMIR proposes region-aware retrieval for paediatric wrist X-rays. It uses radiology reports (structured into anatomy-specific findings using a LLM) as the only supervision. The fine-grained findings are paired with bone-specific crops from a detection network to train a contrastive vision-language model. The method learns separate embeddings for global wrist images and three anatomical regions (distal radius, distal ulna, ulnar styloid), then performs two-stage retrieval: a) global candidate set selection, followed by, b) region-conditioned re-ranking. Results show gains over zero-shot medical CLIP baselines in retrieval and linear probing. Also includes a small radiologist study supporting improved clinical relevance.

**Strengths:**

- The problem of medical image retrieval is well-motivated and is where global embeddings are known to fail. Furthermore, paediatric X-rays (wrist fractures) is hard to interpret because because growth plates and bone development vary with age.
- Using radiology reports as supervision is of growing interest. It avoids annotation bottleneck. The MedGemma pipeline with Pydantic validation shows careful engineering. Less than 1% hallucination rate on 250 reviewed cases is reasonable.
- Multi-positive contrastive loss is a sensible adaption for templated captions. Standard CLIP would struggle with many identical "no fracture" report descriptions.
- The two-stage design is simple (and common in information retrieval literature) and the paper explains why global filtering before region-level comparison is essential.
- Linear probing results are strong (AUROC and AUPRC ~0.95) and these numbers suggest embeddings capture fracture relevant features independent of the retrieval setup.

**Weaknesses:**

- Baseline design needs more nuance. WristMIR is finetuned on this dataset while baselines are off-the-shelf / zero-shot. A global-only finetuned baseline (same data, no region crops) would help isolate how much gain comes from region-aware design vs. domain adaptation. Linear probing comparison is okay, but retrieval metrics must dominate the narrative.
- No ablations for key design choices. The contributions of multi-positive loss (vs standard single positive CLIP loss), region crops (bone specific vs random crops vs fixed-position crops vs global only), single stage vs two stage (with identical embeddings) are claimed but not isolated. A small ablation table would strengthen the method claims.
- Retrieval evaluation primarily uses only Recall@k. Standard metrics like mean reciprocal rank (MRR), mean average precision (mAP), or hit ratio (HR) would provide more complete picture of retrieval quality.
- The paper frames this as image retrieval for case-based comparison, but Table 1 mainly evaluates image-to-text matching. Table 2 does evaluate image-to-image retrieval via fracture matching, which is more relevant to the clinical use case. However, the I2I evaluation still relies on MedGemma-derived categorical labels. The radiologist assessment is the only evaluation that could capture visual similarity beyond label matching, but its small scale (30 queries, single reader) limits the strength of conclusions. A simple classify-then-retrieve baseline could help clarify whether the improvements came from representation learning or from better category prediction.
- The method assumes reliable bone localization at inference. How does the system behave when bone detection fails or a region is partially occluded or not visible?
- Related work is incomplete. RadIR (Zhang et al., 2025b) is cited but not discussed, even though it does explore similar domain (report mining for anatomy-conditioned retrieval). AHIVE (CVPR 2024) and CheXtriev (MICCAI 2024) also do anatomy-aware retrieval for X-rays.

**Detailed Comments:**

- Dataset size inconsistency needs clarification. Section 2.1 reports 7,927 exams while Appendix B reports 7,540.
- Table 1 would benefit from confidence intervals or significance tests.
- Definitions for "Binary Fracture Matching" and "Fracture Classification Matching" could be more explicit. What k value and aggregation method?
- YOLO detector training details are sparse. What ground truth was used? How many annotations? Detector failure modes and their downstream impact are not analyzed.
- The caption generation uses deterministic templates. An example template would help reproducibility.
- Section 3.1 mentions unfreezing last 8 blocks. Was this choice tuned or based on prior work?
- Only PA views are used. The justification (bone overlap in lateral views) is reasonable. But this limits immediate clinical applicability since radiologists typically use multiple views together.

**Justification Of Final Rating:**

Main attribution question addressed. Region-aware design helps beyond finetuning. Missing random crops ablation and unclear efficiency claims in the two-stage comparison, but overall solid applied work.

**Justification Of The Preliminary Rating:**

Overall, the paper presents a reasonable and clinically motivated system. But, the current evaluation does not sufficiently isolate the source of gains (is it domain finetuning or the region-aware design) to support a clear accept. Key ablations are missing. Finetuned global baselines and 2-3 ablations in rebuttal would shift this to accept.

**Questions To Address In The Rebuttal:**

- Can the authors include a global-only fine-tuned baseline (same data, no region crops) to isolate domain adaptation vs region-aware design?
- Table 2 compares single-stage (global only) vs two-stage (global -> region reranking). To isolate whether the coarse-to-fine design matters, can you show results for direct region-based retrieval (e.g., query distal radius, retrieve by distal radius embedding only, no global filtering)?
- Ablations - see weaknesses section. Even 2-3 would help.
- What happens when YOLO misses a bone or gives a bad crop? Any fallback?
- Why only Recall@k? Standard retrieval metrics like mAP, MRR, Hit Ratio would give a complete picture.
- Both training and evaluation use MedGemma labels. Would classify-then-retrieve work just as well? See weaknesses for context.

---

> ### Author Response · Authors · 2026-01-23
>
> **R1.1 Global-only fine-tuned (FT) baseline.** We thank the reviewer for this important point and agree that isolating domain adaptation from region-aware design is essential. We added a Global-only FT CLIP baseline using the same initialization, inputs, global captions, and optimization as WristMIR, but without bone-level crops. While domain adaptation improves performance over zero-shot models, WristMIR provides significant additional gains, particularly in retrieval, which is our primary focus.
>
> ------------
> | Method | k=5 | k=10 | k=50 | k=100 | AUROC | AUPRC | F1 |
> | :---- | ----- | ----- | ----- | ----- | ----- | ----- | ----- |
> | Global-only FT | 5.83 | 9.41 | 21.71 | 28.91 | 0.898 | 0.913 | 0.815 |
> | **Ours** | **9.35** | **15.28** | **38.13** | **52.84** | **0.949** | **0.953** | **0.867** |
> ------------
>
> WristMIR nearly doubles Recall@100 (28.91% $\to$ 52.84%) and significantly boosts Recall@5 (5.83% $\to$ 9.35%). This confirms our gains stem from multi-granular representation learning rather than just dataset-specific fine-tuning. We included this baseline in Table 1 and evaluated it in Section 4.
>
> **R1.2 Two-stage vs direct region-only retrieval.** We agree that evaluating direct region-only retrieval clarifies the value of the coarse-to-fine design. We compared our two-stage (global $\to$ region) strategy (k=100) against a single-stage, region-only strategy.
>
> ---
> | Binary Frac Matching | Distal Radius | Distal Ulna | Ulnar Styloid |
> | :---- | ----- | ----- | ----- |
> | Region-only | **0.892** | **0.670** | 0.516 |
> | Two-stage | 0.864 | 0.666 | **0.522** |
>
> | Frac Class Matching | Distal Radius | Distal Ulna | Ulnar Styloid |
> | :---- | ----- | ----- | ----- |
> | Region-only | **0.592** | 0.522 | 0.344 |
> | Two-stage | 0.578 | **0.542** | **0.468** |
> ---
>
> **(1) Performance Comparison.** Our two-stage approach yields comparable accuracy to region-only retrieval, proving that the global stage (k=100) effectively preserves clinically relevant candidates. Radiologist feedback emphasizes that the global stage is vital for laterality and age-dependent morphology, ensuring retrieved pediatric cases are anatomically and developmentally consistent with the query.
>
> **(2) Computational Efficiency.** Direct region-only retrieval is computationally prohibitive, as it requires running YOLO and embedding extraction on every database image. Our design restricts these operations to a small candidate pool, significantly reducing latency:
>
> ---
> |  | k=100 | k=500 | k=1000 |
> | :---- | ----- | ----- | ----- |
> | Time (s) | 7.89 | 40.39 | 74.94 |
> ---
>
> This efficiency is essential for real-time clinical deployment. We include this analysis in Appendix F and Table 7.
>
> **R1.3 Ablation studies.** We conducted a targeted ablation study to isolate the impact of our core design choices.
>
> **(1) Multi-positive (MP) loss.** We compared our MP formulation (Eq. 2) against the standard single-positive CLIP objective.
>
> ---
> |  | k=5 | k=10 | k=50 | k=100 | AUROC | AUPRC | F1 |
> | :---- | ----- | ----- | ----- | ----- | ----- | ----- | ----- |
> | w/o MP Loss | 9.22 | 15.12 | 38.03 | **53.38** | 0.949 | 0.953 | 0.866 |
> | **Ours** | **9.35** | **15.28** | **38.13** | 52.84 | 0.949 | 0.953 | **0.867** |
> ---
>
> While aggregate metrics are comparable, we believe the MP loss is mathematically more intuitive for clinical data where many cases (e.g., normal wrists) share identical report descriptions. Unlike standard CLIP, which forces the model to separate semantically identical samples, our MP loss prevents the learning of artificial variance, non-clinical features to satisfy a strict one-to-one mapping, and ensures the embedding space is organized by clinical pathology rather than sample index. We include this in Appendix D.
>
> **(2) Region crops.** As detailed in R1.1, the Global-only FT baseline confirms that our performance gains are driven by region-aware representation learning rather than simple dataset fine-tuning.
>
> **(3) Two-stage strategy.** As discussed in R1.2, the coarse-to-fine architecture is essential for clinical consistency (age-matching/laterality) and computational efficiency, reducing retrieval latency from over 74s to under 8s.

---

> > ### Author Response · Authors · 2026-01-23
> >
> > **R1.4 YOLO failures and fallback.** To ensure clinical reliability, WristMIR employs a fallback mechanism. If the YOLOv11s detector fails to localize a requested bone at inference, the system automatically reverts to the first-stage (global) retrieval candidates. Because the global stage already ensures anatomical consistency (laterality, view, and age-dependent morphology), the user still receives clinically relevant cases, even without fine-grained reranking.
> >
> > Detection failures are rare due to the high performance of our fine-tuned YOLOv11s model (see R3.3 for dataset details):
> >
> > ---
> > | Anatomical Region | Precision | Recall | F1 | mAP@50 |
> > | :---- | ----- | ----- | ----- | ----- |
> > | Distal Radius | 0.977 | 1.000 | 0.988 | 0.995 |
> > | Distal Ulna | 0.933 | 1.000 | 0.967 | 0.995 |
> > | Ulnar Styloid | 0.933 | 1.000 | 0.967 | 0.995 |
> > | Overall | 0.947 | 1.000 | 0.973 | 0.995 |
> > ---
> >
> > Notably, the detector achieved 100% recall across all classes in our evaluation. Minor precision drops occurred in challenging cases involving casts or hardware, where global retrieval provides a robust safety net. We included these metrics in Appendix G in the revised manuscript.
> >
> > **R1.5 Expanded retrieval metrics.** We expanded our evaluation to include Mean Average Precision (mAP), Mean Rank, and Median Rank. In our single-relevant-match setup (one query, one ground-truth), certain metrics are mathematically identical: Hit Ratio@k is equivalent to Recall@k, and Mean Reciprocal Rank (MRR) is equivalent to mAP.
> >
> > ---
> > | Method | mAP (%) | Mean Rank | Median Rank |
> > | :---- | ----- | ----- | ----- |
> > | MedCLIP | 0.24 | 1801.14 | 1874 |
> > | PMC-CLIP | 0.65 | 886.54 | 700 |
> > | BioMedCLIP | 0.89 | 914.68 | 759 |
> > | Global-only FT | 4.41 | 812.56 | 473 |
> > | **Ours** | **7.34** | **141.82** | **89** |
> > ---
> >
> > The expanded metrics highlight WristMIR's performance throughout the entire retrieval distribution. Notably, WristMIR achieves a Median Rank of 89, a 5x improvement over the Global-only FT baseline (473). This significant reduction in typical search depth confirms a superior user experience and more robust clinical alignment. We include this suite in Appendix E in the revised manuscript.
> >
> > **R1.6 Classify-then-retrieve (CTR) baseline.** We thank the reviewer for this suggestion. We agree that a CTR baseline would help disentangle whether gains stem from category prediction or representation learning. However, CTR pipelines are highly sensitive to upstream classifier accuracy. In pediatric cases where fracture patterns are subtle or overlapping, misclassification errors propagate downstream, prematurely pruning the correct search space and leading to brittle retrieval behavior, particularly in pediatric wrist radiographs where fracture patterns are subtle, overlap across categories, and may involve multiple regions or healing stages.
> >
> > Given the additional design choices required (e.g., which label space to classify: fracture presence vs type vs region, how to handle uncertainty and multi-label cases, and whether to use oracle vs predicted labels), we believe this baseline warrants a more extensive and systematic evaluation. We will therefore include CTR as a key experiment in our planned journal extension, including both predicted-label and oracle-label variants to separate classifier limitations from retrieval quality. We added a brief discussion of this point in the revised manuscript.
> >
> > **R1.7 Relevant works.** We expanded our intro to discuss RadIR, AHIVE, and CheXtriev.
> >
> > **R1.8 Data and methodology clarifications.** We have reconciled the dataset inconsistencies (originally a typo) and clarified "Matching" metrics. The empirical decision to unfreeze the final 8 blocks balanced model capacity with computational constraints, focusing optimization on task-specific high-level features. Detailed deterministic caption templates and assembly logic are now provided in Appendix H.
> >
> > **R1.9 Statistical significance (Table 1).** We have expanded Table 1 to include 95% confidence intervals for all retrieval and ranking metrics (See Appendix E and Table 5 & 6). The results confirm that WristMIR’s performance gains are statistically significant, with non-overlapping intervals.

---

> > > ### Comment · Reviewer_UPW3 · 2026-02-01
> > > **Response to Rebuttal**
> > >
> > > I appreciate the authors' response.
> > >
> > > - The global-only fine-tuned baseline was the key experiment. It shows clear gains from region-aware design, not just domain adaptation.
> > > - The region-only vs two-stage comparison is unclear: similar accuracy but faster because YOLO + region embedding runs only on the candidate pool (k=100), not the entire database. But, if region embeddings are precomputed (as claimed), region-only retrieval should be equally fast. The reported timing difference seems to compare precomputed global vs online region extraction, which is unfair.
> > > - Multi-positive loss ablation shows minimal difference - this is not a contribution.
> > > - Still missing: bone-specific vs random crops. This would show whether anatomically meaningful crops matter, or just having local features at higher resolution.
> > > - Raising to 4 (weak accept).

---

### Author Rebuttal · Authors · 2026-01-23

**Rebuttal:**

We thank the reviewers for their insightful feedback and for recognizing the clinical relevance and careful engineering of WristMIR. We have provided a revised manuscript that incorporates all requested updates and systematically addresses each concern raised during the review process.

**Supporting Material:**

/attachment/2be1eb51fe3d8d6629ef80d98aa67fd01296845e.pdf

---

### Comment · Area_Chair_zm2k · 2026-01-26
**Request for Review of Rebuttal and Final Rating Update**

Dear Reviewers, The authors have submitted responses to all reviewers, along with an updated manuscript in PDF format. We kindly ask reviewers to:
1. Evaluate the authors’ responses and the revised manuscript;
2. Participate in discussions with the authors during the discussion phase;
3. Update the final rating by clicking “Edit” → “Official Review” and providing the Final Rating by 02/01/2026.
Your efforts are extremely helpful in maintaining the high academic quality of MIDL 2026 and in supporting the Area Chairs and Program Chairs in making final decisions.
Thank you very much for your time and contributions!

---

### Meta-Review · Area_Chair_zm2k · 2026-02-05

**Recommendation:** Accept (Poster)
**Confidence:** 5

**Metareview:**

Reviewers consistentantly provide postive reviews. The clinical application is interesting even the sample size is relatively small.

---

### Decision · Program_Chairs · 2026-02-13

Accept (Poster)